# Religious Policy of the Mamluk Sultan Baybars (1260–1277 AC)

Hatim Muhammad Mahamid 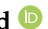

Academic College for Teacher Education, The College of Sakhnin, Sakhnin 3081000, Israel;
hatim_mahamid@hotmail.com

**Abstract:** This study focuses on the religious policy of the Mamluk Sultan Rukn al-Din Baybars (d. 1277), and its application throughout his rule in Egypt and Syria (Bilād al-Shām). This study also discusses the impact of this policy and its benefit for Muslims and Islam in general in the region. Dealing with the character of Baybars of Mamluk origin is very important in this study to obtain the required answers to the questions related to the Islamic character of Mamluk rule in the regions of the Middle East in the Middle Ages. Although Baybars' background was as a slave (*mamlūk*), who was not familiar with the religion of Islam, he had to prove his ability to act as a leader of an Islamic state. Baybars worked to implement the principles of Islam according to the Quran and the tradition of the Prophet (*sunna*), thus exploiting his status as an Islamic leader and as Sultan during his reign. Baybars was strict in his religious policy toward those who committed forbidden acts according to the Islamic religion, and he was keen to punish them and implement the *Sharīʿa* as required, in cooperation with the ʿ*ulamā*. Baybars was given relevant degrees and titles he deserved by the revived Abbasid Caliph in Cairo, such as a "holy warrior" (*Mujāhid*), "partner of emir of believers/the Caliph" (*Qasīm Amīr al-Muʾminīn*), and other religious titles that testify his high-ranking religious status as defender of Islam.

**Keywords:** religious policy (*al-Siyāsa al-Sharʿiyya*); Islamic religion; Sultan Baybars; Mamluk era

## 1. Introduction

During the reign of the Mamluk Sultan Baybars (1260–1277 AC), he worked in various religious and political fields (*sharīʿa and siyāsa*) until he succeeded in achieving stability, security, and a strong system of government. After the revival of the Abbasid caliphate in Cairo, Baybars was granted high titles that symbolized religious status as well as political and military status through documents issued by the caliph or other decrees (see: Ibn Taghrī Birdī 1963, vol. 7, pp. 111–13; al-Maqrīzī 1997, vol. 1, pp. 531–34, 547–48; ibid, vol. 2, pp. 22–24; al-Nuwayrī 2004, vol. 30, pp. 174–79).

Most of historians and chroniclers agree that Baybars' background was as a slave (*mamlūk*) of Turk origin, with a lack of Arabic language and Islamic knowledge (ʿ*ilm*). But throughout his reign (1260–1277), he proved to be considered as an Islamic leader by implementing the Islamic principles of different aspects according to the Quran and the Prophet's tradition (Ibn ʿAbd al-Ẓāhir 1976; Ibn Shaddād 1983, vol. 31). Several studies dealt with Sultan Baybars' policy and arrangements, regarding the legitimization of his power, and how he sought to compensate for his lack of dynasty by using other strategies (see: Troadec 2014–2015, pp. 113–47; Van Steenbergen 2015, pp. 1–44; Mahamid 2023). Baybars was described by the chronicler Ibn ʿAbd al-Ẓāhir (1976, p. 225) as follows: "…He spent his time serving the interests of Muslims and taking care of matters of religion".

Sultan Baybars faced many challenges that led to a controversial plea regarding legality in his position, accompanied by fierce military confrontation. In his study about Mamluk religious policy, Berkey (2009, pp. 7–22) argues that the relations between political and religious authority in Islam are much more complex. Baybars continued his holy wars (*jihād*) fighting against the Franks and the Mongols, in addition to suppressing and facing threats to existence to the Islamic State, including external and internal threats. Otherwise,

Baybars acted on the unification of the country under his authority and implemented Islamic laws, contributions, and reforms in different religious aspects, such as reviving the Abbasid Caliphate in Cairo and judicial reforms. The Mamluk State, then, had been expanded and unified from the south of Egypt to North Syria, including Yemen and the most holy places of Islam in Hijaz and Palestine.

Berkey begins his study about Mamluk religious policy by stating an Arabic saying "*Al-Islām dawlah wa-dīn*" (Islam is a state and a religion), that developed with the time. He argues that the relation between political and religious authority in Islam is much more complex (Berkey 2009, pp. 7–22). Contemporary scholars specializing in the Mamluk era, such as David Ayalon (1994), focused their studies on the Mamluks as powerful warriors and foreigners in general, giving their rule the image of military rulers, with less interest in social and even religious matters. Mauder (2021, pp. 80–111) critically reconsiders the idea of the Mamluk rulers' lack of interest religious affairs and the power to which they could confer supreme religious status. This is performed by focusing on stating the example of the last Mamluk sultan in Egypt, Qānṣwa al-Ghawrī (r. 906/1501-922/1516). The sultan al-Ghawrī used good relations and processes of capital exchange to bind scholars who possessed skills to his court, which could be helpful in his project of legitimizing Mamluk sultanic authority. He had strengthened his religious authority through conducting councils of religious sciences (*majālis al-ʿilm*) in his court and participating in processes of the transmission of knowledge (ʿAzzām 2012, pp. 44–45).

This study examines Baybars' religious policy during his rule, and the ways he acted to implement his religious policy (*al-siyāsa al-sharʿiyya*). This study also aims to follow the process of achieving Baybars' goals and the extent of forming a unified authority of religious and political images, for which he paved the way for the Mamluk rulers and sultans to follow his religious policy after him. To reach the desired results of this study in the most appropriate way for this type of research, it relies on the literary research method regarding the personality and works of Sultan Baybars, while comparing and analyzing his characteristics and approach to governance and the application of the practices of the Islamic religion.

Two important chroniclers of Baybars' era provide a detailed description of the biography of Sultan Baybars and the period of his rule, Ibn ʿAbd al-Ẓāhir (d. 1292) and ʿIzz al-Dīn Ibn Shaddād (d. 1285), which reveals a clear picture of his rule and deeds. This study also relies on conclusions and extrapolations of Baybars' personality through official documents and letters received from several sources, including religious, military, and foreign sources, as well as through letters and decrees issued by him. This research also attempts to compare with the propositions and conclusions of the modern historical literature and research that specializes in Sultan Baybars and the Mamluk era.

To examine Baybars' religious policy of strengthening the Islamic character of the state, several key questions are discussed: What qualities and characteristics of religious knowledge and authority helped Baybars face the challenges threatening the Islamic reality that time? What steps and methods have been taken in implementing his Islamic goals and reforms? Did Baybars succeed in implementing Islamic policy during his tenure as Sultan? What is the degree of change in the religious character of the Mamluk state in his time?

## 2. Applying and Activating the Religious Policy of Baybars

### 2.1. Between Baybars and ʿUlama: Reforms in Juridical System

As mentioned by historians of the Mamluk era, Sultan Baybars was interested and eager to understand the issues of the Islamic religion and work to implement it. Ibn Taghrī Birdī (1963, vol. 7, p. 219) mentions that Baybars was distinguished by his intelligence and activity, his benevolence to the poor, honoring ʿulama, and listening to their advice, as he used to do with the famous ʿulama like al-ʿIzz ibn ʿAbd al-Salām (d. 1262) and Abū Zakariyyā Yaḥyā al-Nawawī (d. 1277). In his travels, Sultan Baybars used to be accompanied by many senior officials and ʿulama, such as the Ḥanafi judge, Ṣadr al-Dīn Sulaymān bin ʿAbd al-Ḥaqq, in the year 667 H/1269 AC, when the Sultan traveled to perform the rituals of

*Ḥajj*. Baybars used to ask the judge religious questions to understand the Islamic religion and widen his knowledge (see: Ibn ʿAbd al-Ẓāhir 1976, pp. 354–58; al-Maqrīzī 1997, vol. 2, pp. 60–62; al-Nuwayrī 2004, vol. 30, pp. 107–8).

Baybars was interested in the work of judges and the integrity of their work in establishing *sharīʿa* and justice throughout his domain and intervened in religious administration by dismissing and appointing judges. In his study, Lev and other researchers state that the relations between the Mamluk state and ʿulama were symbiotic, which helped them rule and endowed their regime with its Islamic content (Lev 2009, pp. 1–26; Berkey 2009, pp. 7–22). In 659 H/1261 AC, for example, Baybars dismissed Judge Badr al-Dīn al-Sinjāri, appointed Ibn Bint al-Aʿazz in the position of the main judge (*qāḍī al-qudāt*) in Egypt, and delegated to him the supervision of the *waqf* affairs and mosques (Ibn ʿAbd al-Ẓāhir 1976, p. 84; al-Maqrīzī 1997, vol. 1, p. 528; al-Nuwayrī 2004, vol. 30, pp. 7–9; Jackson 1995, pp. 52–65). In the same year, Judge Shams al-Dīn Ibn Khallikān was appointed in Syria (*al-Shām*), where he was delegated to rule from al-ʿArīsh and the Euphrates, and he was responsible for preserving the *waqf*s of the mosque in Damascus, the *Maristān,* and teaching in seven main *madrasa*s (Ibn Shaddād 1983, vol. 31, pp. 274–75; al-Maqrīzī 1997, vol. 1, pp. 538–39, 44; al-Dhahabī 1999, vol. 48, p. 75; al-Nuwayrī 2004, vol. 30, p. 28). Here I argue that the purpose of the new quadruple structure of the judiciary was two-fold: to create a uniform but at the same time flexible legal system. The need for predictable and stable legal rules was addressed by limiting qāḍīs' discretion and promoting taqlīd, i.e., adherence to established school doctrine.

As a significant act, in 661/1263, Sultan Baybars ordered for a judicial reform in Egypt by appointing four judges from the four Islamic schools of law (*madhāhib*). Some studies deal with the purpose of the new reform of the judiciary by Sultan Baybars aiming to create uniformity in the religious system but with flexibility in promoting the imitation of Islamic tradition (*taqlīd*) (Nielsen 1984, pp. 167–76; Rapoport 2003, pp. 210–28; Mahamid 2023). At first, in 662/1264, Baybars appointed vice judges as deputies of the main Shafiʿi judge of Egypt, Taj al-Dīn Ibn Bint al-Aʿazz. al-Maqrīzī (1997, vol. 1, pp. 544, 562), regarding this event, said: "This was not known in Egypt before this time". In the following year, 663/1265, the Sultan saw the necessity of applying *sharīʿa* to the diversity of jurisprudence due to the controversy and differences in judgements between the four schools of thought. So, Baybars established the appointment of the four judges of the four *madhhab*s, with each of them in the position of "*qāḍī al-qudāt*" (judge of judges) of his school in Egypt, and that everyone should judge according to his *madhhab* and independently take their responsibility to the rest of the Egyptian regions (Escovitz 1984, p. 23; Jackson 1995, pp. 52–65). This way, Baybars had treated the four schools of law equally. The first four independent judges were Tāj al-Dīn Ibn Bint al-Aʿazz for the Shafiʿis, Ṣadr al-Dīn al-Adhruʿī for the Hanafis, Sharaf al-Dīn ʿUmar al-Subkī for the Malikis, and Judge Shams al-Dīn Muḥammad ibn Ibrāhīm for the Hanbalis. Medieval Muslim historians mention that the reason for appointing the four judges in Egypt was that the Shafiʿi judge Taj al-Dīn Ibn Bint al-Aʿazz had stopped implementing many rulings of the other *madhāhib*, so many complaints were frequent, and many affairs were disrupted. Thus, a favorite emir of Baybars, Jamal al-Dīn Aydughdī, served four judges in cooperation with the Sultan (Ibn Taghrī Birdī 1963, vol. 7, pp. 121–22; al-Maqrīzī 1997, vol. 2, pp. 27–28; al-Dhahabī 1999, vol. 49, p. 21; al-Nuwayrī 2004, vol. 30, pp. 75–79, 93–94).

Similarly, in Damascus, Sultan Baybars expanded his reforms in judicial matters in Muharram 664/October 1265, when three other judges from the other *madhāhib* were appointed, in addition to the Shafiʿi Judge Shams al-Dīn Ibn Khallikān. From the accounts of Mamluk sources, it seems that in the beginning, the Judges of the Malikis and Hanbalis in Damascus hesitated to accept this post, but then, they agreed after an obligatory decree issued by Sultan Baybars. Therefore, this reform had been performed throughout the other parts of the Mamluk provinces in Egypt and Syria, and it continued to become the norm (Ibn Taghrī Birdī 1963, vol. 7, p. 137; Ibn Shaddād 1983, vol. 31, pp. 235–38; al-Maqrīzī 1997, vol. 2, pp. 31–32; al-Nuwayrī 2004, vol. 30, p. 79).

*2.2. Dār al-ʿAdl: Implementing Justice in Accordance with Religion*

During the unrest prevailing throughout the Ayyubid and Mamluks, Sultan Baybars was interested in building the House of Justice (*dār al-ʿAdl*) in Cairo and paying attention to its work to spread justice and safety among people and officials. In his study, Nasser Rabbat (1995, pp. 3–28) deals with Baybars' *Dār al-ʿAdl* within the administration of justice in general, with its fundamental purpose of the Islamic state. Baybars had inaugurated the *dār al-ʿAdl* in 661–662/1263–1264, where he himself sat down to justice. This can be deduced from the study of Rapoport (2012, pp. 71–102) who focused extensively on royal justice and religious law (*siyāsa and sharīʿa*) under the Mamluks. The Sultan aimed to ensure justice and help poor people by distributing the poor people and needy to emirs and other troops in Cairo to support them with food and help, because of the difficult situation of high prices (Ibn ʿAbd al-Ẓāhir 1976, pp. 182–83, 188–90). Ibn Kathīr (1988, vol. 13, p. 234) mentions an example as a model of justice, that Baybars himself stood trial in front of the main judge Ibn Bint al-Aʿazz in 660/1261, although he was right and not guilty. The sit-in of Sultan Baybars in *dār al-ʿAdl* for justice was an approach that he used to judge people fairly and look at the affairs of the public, with his presence in Dār al-ʿAdl in the month Dhū al-Ḥijja 671/1273, calling and encouraging poor and oppressed people to demand their rights. He also did the same in *Muḥarram* 672/July 1273, to treat the oppressed with justice and rescue the rights for the sake of the poor (Ibn Taghrī Birdī 1963, vol. 7, p. 163). Ibn Shaddād (1983, vol. 31, pp. 61, 70, 277–82) mentions examples of seeking justice by Baybars, not only for oppressed Muslims, but also for Jews and Christians (*ahl al-dhimma*). He praises Baybars' manners in such cases of seeking justice by stating a Prophet *hadith*: "justice of a day is equal to forty years of worship" (*ʿadlu yawm yaʿdilu ʿibādat arbaʿīn sana*). In the same sense, Ibn Kathīr (1988, vol. 13, p. 223) fairly describes Baybars' rule with justice in appointing and dismissing people, and he was sufficiently courageous and brave that God sent him to people who needed him at this difficult time.

The Mamluk sultans cared about looking into grievances "*maẓālim*" and dealing with their effects among people. Fuess (2009, pp. 121–47) dealt in his study with the policy of "*maẓālim*" by the Mamluks in general, and the political implication of using the *maẓālim* Jurisdiction by the Mamluk Sultans, while others studied royal justice in Mamluk Cairo where the Mamluks in general took their judicial responsibilities quite seriously (Petry 1994, pp. 197–211; Berkey 2009, p. 15). Baybars himself tried to be a role model in the application of justice, when he was concerned with applying and preserving Islamic law, as he stood before the judge to confront the prosecution to prove his equality with the rest of the people in the law of *sharīʿa* and to establish the sanctity of religion (Ibn ʿAbd al-Ẓāhir 1976, pp. 84–86). The presence of Sultan Baybars at the Justice House in Damascus in 673/1275 testifies his seeking justice after he had issued an order to confiscate properties and orchards. This presence aimed to present proof for his action before the four judges and *ulama*. As a result, the Sultan responded to the judges' decision, and his order was cancelled (Ibn Taghrī Birdī 1963, vol. 7, pp. 246–47).

Baybars was characterized by his clarity to anyone who addressed him with a complaint or raised a grievance to him, as well as his justice in returning rights to their owners "*radd al-amānāt ilā ahlihā*". Baybars' act with the main Hanbali judge in Egypt, Shams al-Dīn Muḥammad al-Maqdisī, in 670/1272, testified his policy in this aspect. Baybars ordered to restore deposits and properties that the judge had seized for merchants from Baghdad, Harran, and al-Sham (Syria). In this incident, al-Maqrīzī says "... the judge was re-arrested in the castle, and he held a two-year prison sentence, and the Sultan did not appoint anyone of the Hanbalis after him" (Ibn Shaddād 1983, vol. 31, pp. 31–32; al-Maqrīzī 1997, vol. 2, pp. 78–79). Medieval sources mention various facts and examples of good treatment and the follow-up of Sultan Baybars in revealing the truth and returning the rights to their owners, as an example of returning the property and deposits of a Jewish merchant in Homs (Syria) (Ibn Taghrī Birdī 1963, vol. 7, pp. 180–81; Ibn ʿAbd al-Ẓāhir 1976, pp. 77–79; al-Nuwayrī 2004, vol. 30, pp. 122–23).

*2.3. Baybars' Policy of Commanding Right and Forbidding Wrong*

One of the demands and principles of the Islamic religion and its interest in human ethics and religious obligations, as stated in the Qur'an and Sunnah, is enjoining good and forbidding evil (*al-amr bil-ma'rūf wal-nahy 'an al-munkar*). In his study, Cook (2010) attempts a comprehensive analysis in an attempt to map the history of Islamic thought and reflection on "commanding right and forbidding wrong", and to explain its relevance for politics in the Islamic world in general.

Regarding Sultan Baybars, he worked hard at preserving public morals within the framework of Islamic law and acted strictly against those who contradicted religion (*sharī'a*). The transitional period between the Ayyubid and Mamluk rule influenced the spread of evil and weakness of religion, in addition to political crisis. Al-Dhahabi notes this matter by saying "The religion was weak during the reign of the Ayyubid King al-Nāṣir (d. 659/1260), with wine and adultery, spread of injustice and emergence of innovations (*al-bida'*) and other negative acts" (al-Dhahabī 1999, vol. 48, pp. 29, 31–32).

Mamluk sources mention various events of Baybars, in which he acted against denied behaviors. In 661/1263, for example, the Sultan sat at the *dār al-'adl* (justice house) and ordered the cleaning of Alexandria from the denied behaviors of the Franks, which were prevalent in that area (Ibn 'Abd al-Ẓāhir 1976, p. 176; al-Maqrīzī 1997, vol. 1, p. 560). Baybars' act in fighting against such a phenomenon has widened in Cairo itself, where he supervised people at night and held accountable those who carry out such behaviors from some of the Mamluks, such as deputies, governors, commanders, and others. Al-Nuwayrī and al-Maqrīzī note an incident, in which Baybars himself acted against such a phenomenon in the month *Dhū al-Ḥijja* 663/1265, in which the Sultan left the castle and wandered in Cairo at night disguised to control the conditions of the people. When he saw some of the Mamluks doing disgraceful acts without fear of religion or morality, he ordered them to be held accountable as punishment by cutting off the hands of some of them. Furthermore, Baybars also came out strongly against singing and dancing, and he is quoted in the Mamluk sources as announcing against such denied acts, such as acting against drinking wine and other disgraceful actions, as his order regarding spoiling wine in his domain in 669/1271 and the death penalty for those who do it (Ibn Taghrī Birdī 1963, vol. 7, p. 154; al-Maqrīzī 1997, vol. 2, pp. 28, 34, 59, 100; al-Dhahabī 1999, vol. 49, pp. 59–60; al-Nuwayrī 2004, vol. 30, pp. 72, 117).

Baybars saw that his duty to expand the fight against deniable phenomena, not only in Egypt, but in all countries under his domain. It seems that the policy of Sultan Baybars and his orders to remove wine and acts of corruption had been gradually achieved until it was implemented not only in Cairo, but in all regions of Egypt and other countries. Mamluk sources mention in the events of 664/1266 that the sultan increased his denial of evil, and he ordered spoiling wine and excluding effects of evil. So, bars in all his kingdoms in Egypt and Syria were banned, forbidding wine and cannabis (*ḥashīsh*). Baybars stressed the punishment of the wine users, which led him to call the people on the feast days of the same year: "Whoever drinks wine or brings it will be hanged". It is also mentioned in the events of 670/1272 that Sultan Baybars was firm in fighting against corruption and evil. He continued to emphasize the spilling of alcohol and the removal of negative acts. He also ordered the cancellation of wine and evil when he came to Homs in Syria in 666/1268, and the process of revoking wine in Damascus in 668/1270 (al-Maqrīzī 1997, vol. 2, pp. 38, 49, 75; al-Dhahabī 1999, vol. 49, pp. 50–51). In the same year, after the conquest of Safed, the Sultan prevented the trading of cannabis (*ḥashīsh*) and ordered the discipline of those who used or traded it. Cannabis use was common at that time in Syria and other places, especially near the coastal areas adjacent to areas of the Franks' influence (Ibn 'Abd al-Ẓāhir 1976, p. 266; al-Maqrīzī 1997, vol. 2, p. 36; al-Dhahabī 1999, pp. 48, 20; al-Nuwayrī 2004, vol. 30, p. 84).

Baybars was strict in prosecuting those who deal in such corrupt forms, even those in high positions, officials or 'ulama, as he did with the "sheikh of the Sultan" Khadr ibn Abi Bakr, in accusing him of acts of evil and committing corruption. So, the sheikh was

arrested and imprisoned in the castle of Cairo in 671/1273 (Ibn Shaddād 1983, vol. 31, pp. 58–60; al-Maqrīzī 1997, vol. 2, p. 82; al-Nuwayrī 2004, vol. 30, pp. 127–28). This severe policy of Baybars regarding such bad activities had become firmly applied, and in 674/1276, the Sultan ordered the hanging of a high-ranking Mamluk emir, Shujāʿ al-Dīn ʿAnbar, because of drinking wine, and hung him over the castle (Ibn Shaddād 1983, vol. 31, p. 133; al-Maqrīzī 1997, vol. 2, p. 95). Sultan Baybars' emphasis on fighting evils that contradicted Islamic law has increased. He had also banned musical performances of fun and singing, except military activities, such as exercises, military shows, or hunting trips. He even used to serve charity to all other communities except singers and entertainers throughout Sultan Baybars' era. An example of the ceremony of circumcision of Sultan Baybars' son, prince Najm al-Dīn Khaḍr, in 672/1274, testifies that Baybers applied this policy on himself, that the ceremony was modest, and the Sultan did not accept any gift from others (al-Maqrīzī 1997, vol. 2, pp. 86–87).

Medieval historians quote the warm greeting with which Baybars had been welcomed in the Seljuk Qaysary in 675/1277, in which he prevented the use of singing, musical instruments, and entertainers in his reception in the palace of the Seljuk King Ghayyāth al-Dīn, saying to them: "This form does not fit us, and this is not the place of singing, but the place of thanks" (*hādhihi al-hayʾa lā tattafiq ʿindanā, wa-mā hādhā mawḍiʿ al-ghināʾ, bal mawḍiʿ al-shukr*). Therefore, the people there, ʿulama, preachers, officials, notables, women, and children, welcomed him with joy, and the Sufis celebrated him with religious praises (*dhikr*), chants, and hymns (Ibn ʿAbd al-Ẓāhir 1976, pp. 465–67; al-Maqrīzī 1997, vol. 2, p. 100; al-Nuwayrī 2004, vol. 30, pp. 228–29).

### 3. Baybars' Religious Contributions: *Waqf* and Institutions

#### 3.1. Maintenance of the Holy Places in Hijaz (al-Ḥaramayn al-Sharīfayn)

One of the main duties of an Islamic leader is to control, protect, and preserve the Islamic holy places for promoting and implementing free religious prayers and visits. Since 652/1254, following the political vacuum and power struggles between the Ayyubids and the Mamluks, in Egypt and Syria, the dispute intensified between the princes of Mecca (*umarā*) over control of the city and the holy sites, which negatively affected the pilgrimage season and holy visits, and the security and safety of the pilgrims (al-Maqrīzī 1997, vol. 1, pp. 481, 486, 487, 491; al-Dhahabī 1999, vol. 48, pp. 14, 16; Ibid vol. 49, p. 53; Ibid vol. 50, p. 23; al-Nuwayrī 2004, vol. 30, p. 95). When Baybars became Sultan, he was interested in strengthening his position as protector of Islam and its holy places in the face of external and internal dangers. He extended his influence and authority to the Hijaz, which contained the two main Holy Sites (*al-Ḥaramayn al-Sharīfayn*). As a first step, from the early days of his rule, Sultan Baybars sent necessary equipment, funds, and items in 659/1261, with emir ʿAlam al-Dīn al-Yaghmūrī for repairing the Prophet's Mosque in Medina. The Ayyubid King al-Mughīth ʿUmar ibn al-ʿĀdil, the governor of Karak, who was the antagonist of Sultan Baybars, was an obstacle in the way to Hijaz. Thus, Baybars had to get rid of him, so King al-Mughīth was arrested and killed in 661/1263 and Karak fell under Baybars' rule (al-Maqrīzī 1997, vol. 1, pp. 526–27, 556–57; al-Nuwayrī 2004, vol. 30, p. 10).

Various measures were taken by Sultan Baybars and other Mamluks against violators of the order of the holy places in Mecca and Medina, and against the Bedouins along the pilgrimage routes (Mahamid 2023). Through their study on "Rakb al-Ḥajj al Shāmī" and the relations between nomads and Pilgrims, Mahamid and Nissim (2023, pp. 307–10) concluded that Sultan Baybars acted severely against the Bedouins and emirs of Mecca, taking special care to secure the pilgrimage routes to the holy places in Mecca and Medina. Furthermore, Walker (2009, pp. 83–105) also discussed the relations between Mamluk authorities and the Bedouin tribes in Transjordan and its tasks, including the defense against foreign invasion and preserving security on the main ḥajj route from Damascus to Mecca.

In 662/1264, the Sultan saw the importance of the Hijaz country and its ways from Syria to Yemen, that, after the conquest of Karak, Baybars obliged the Bedouin tribes to

guard the roads to Hijaz (Ibn ʿAbd al-Ẓāhir 1976, pp. 165, 220; al-Maqrīzī 1997, vol. 1, p. 557; al-Nuwayrī 2004, vol. 30, p. 152). In a sermon delivered to Sultan Baybars in Mecca in 662/1264, the sultans' secretary Jamāl al-Dīn Ḥusayn Ibn al-Mūṣilī received the key of the Kaʿba from the Sultan, indicating that the holy places were protected under the control of the Mamluk state in Egypt. In celebration of this event, visits to people in the *Kaʿba* were allowed for three days without payments (al-Maqrīzī 1997, vol. 2, p. 4; Ibn ʿAbd al-Ẓāhir 1976, pp. 89–90, 183–84).

Therefore, Sultan Baybars was keen on preserving the sanctities, so he sent a group of builders, craftsmen, wood, and other machinery to plan the reconstruction of the Prophet's Mosque. The *kiswa* (cover of the *Kaʿba*) was performed on the custom with a special celebration in Cairo before leaving to the Hijaz, which became a custom of the Sultan every year, accompanied by officials, judges, ʿulama, Quran readers, Sufis, preachers, and others (Ibn ʿAbd al-Ẓāhir 1976, p. 200; al-Maqrīzī 1997, vol. 1, p. 562; ibid, vol. 2, pp. 9, 60; al-Nuwayrī 2004, vol. 30, p. 74). In addition to the maintenance, restoration, and repairs of the holy sites in the Hijaz, Sultan Beybars tended on the pilgrimage mission (*maḥmal*), money and grain to the Hijaz. In *Ṣafar* (664/November 1265), Baybars sent with the deputy of the House of Justice an amount of ten thousand dirhams and salaries of professions to repair the Prophet's Mosque (Ibn ʿAbd al-Ẓāhir 1976, p. 247; al-Maqrīzī 1997, vol. 2, p. 32).

In addition to promotion and control of the holy sites, Baybars organized the administration and securing of roads to perform the religious rituals of pilgrimage and other religious visits (*ḥajj* and ʿumra). Sultan Baybars took care of the safe management of the holy places by distributing powers in Medina and making peace between the conflicting parties over the emirate's position. In *Ramadan* of 665/1267, for example, the decree of appointment was equally written between the opponents of the emirate and the management of the endowments of Medina was arranged in Syria and Egypt (Ibn Taghrī Birdī 1963, vol. 7, p. 146; al-Maqrīzī 1997, vol. 2, p. 44).

As well as in the management of affairs and reform in Mecca, there was a dispute between the princes of Mecca Sharīf Najm al-Dīn Abī Numayy and his uncle Sharīf Bahāʾ al-Dīn Idrīs. In *Shaʿbān* 667/1269, after the reconciliation between them, the Sultan arranged for them twenty thousand dirhams a year and did not take taxes in Mecca as an act of charity "*wa-sabbal al-Bayt al-Sharīf li-sāʾir al-nās*", no one was prevented from visiting the *Kaʿba*, and no one was allowed to devise plots to merchants. So, as a sign of the Sultan's authority in Mecca, coins had been made with the Sultan's name, and preaches would be mentioned in prayers. Then, the Sultan issued a decree of the appointment in the emirate of Mecca, giving the two deputies (*emirs*) responsibility in managing the *waqf* affairs belonging to the *Kaʿba* in Egypt and Syria. In the events of 667/1269, the sources described Baybars' travel to Mecca for pilgrimage, in which he presented charity to poor people, appointed other deputies to help the two emirs of Mecca, and increased the amount of money and crops given to the emirs every year, and he was also generous with the emir of Yanbuʿ and other high-ranking persons in Hijaz (Ibn ʿAbd al-Ẓāhir 1976, pp. 359–60; al-Maqrīzī 1997, vol. 2, pp. 59–62; al-Nuwayrī 2004, vol. 30, pp. 107–08). It seems that making the *Kaʿba* (*al-Bayt al-Ḥarām*) in Mecca a charity (*tasbīl*) was intended to facilitate the visit in pilgrimages of the *ḥajj* and ʿumra, and reduce the expenses incurred by visitors from paying the excise. al-Maqrīzī (1997, vol. 2, p. 95) mentions, for example, the events of 674/1276, saying "the pilgrims of Egypt stayed in Mecca eighteen days, and in the Prophet's city (Medina) ten days, that was not happened before".

### 3.2. Endowments and Charity of Baybars in al-Quds (Jerusalem) and al-Khalīl (Hebron)

Jerusalem and Hebron, as sacred places for Muslims, are part of the interest of the Mamluk sultans in general and the interest of Sultan Baybars in particular, in various aspects of life, especially the religious one. Therefore, various studies have dealt with such religious topics (Mahamid 2003, pp. 329–54; 2013, pp. 73–74; Mishʿal 2011, pp. 63–85; Frenkel 2001, pp. 153–70; Holt 1980, pp. 27–35). Sultan Baybars used to visit Jerusalem and Hebron during his travel between Egypt and Syria. He noticed that the holy sites there

were faded and withered; so, he sent manufacturers and tools to the architects of the Dome of the Rock in Jerusalem. Then, he took out what was with the emirs from the endowments (*waqf*) of Hebron, and added an adjacent village known as Idhnā as *waqf* for the holy site in Hebron (al-Maqrīzī 1997, vol. 1, pp. 526–27; al-Nuwayrī 2004, vol. 30, p. 10). Furthermore, in 661/1263, Sultan Baybars himself traveled to Jerusalem and inspected the conditions there. He saw the need of the mosque for constructions and repair, so he supervised the *waqf* and acted to preserve it, and arranged the interests of the mosque, with the pay of five thousand dirhams each year (Ibn ʿAbd al-Ẓāhir 1976, p. 162; al-Maqrīzī 1997, vol. 1, p. 556).

Several studies dealt with the Mamluk contributions of *waqf*s and institutions as religious and political aims, that helped in reshaping and designing a new sacred topography in Syria and the surrounding areas. Thus, Baybars placed the foundation for reviving the city life in Jerusalem after a period of neglect, and other Mamluk leaders imitated him. It is noticeable that most of Mamluk sultans and high-ranking Mamluk officials left their impact on the city in various aspects of life (Luz 2002, pp. 133–54; Mahamid 2003, pp. 329–54; 2013, pp. 156–71; 2023; Amitai 2017, pp. 156–86).

To encourage visits (*ziyāra*) to Jerusalem and Hebron, the Sultan dedicated several villages adjacent to Jerusalem and in Syria in 662/1264 to spend their proceeds on bread and other expenses for the visitors of Jerusalem. In addition, Baybars ordered several constructions in the city, for commercial and public use, such as an inn (*khān*), bakery (*furn*), and a mill (*ṭāḥūn*), that were administered by the emir Jamāl al-Dīn Muḥammad b. Nahar (al-Maqrīzī 1997, vol. 2, p. 14). In 663/1265, the *waqf* deed of Baybars' *Khan* (inn) in Jerusalem was read at the Sultan's Majlis in the Cairo Citadel, by the chief judge, Ibn bint al-Aʿazz. In the same year, after the "*simāṭ al-Khalīl*" (charity of presenting food for poor and visitors in Hebron) had been canceled for years, Baybars rearranged it with the payment of salaries for the residents and visitors of the holy sites in Hebron (Ibn ʿAbd al-Ẓāhir 1976, pp. 89–90, 183–84, 220–21; al-Maqrīzī 1997, vol. 2, p. 4; al-Nuwayrī 2004, vol. 30, p. 71).

Sultan Baybars used to visit Jerusalem and Hebron very often to verify their situation or as a religious visit. In 664/1266, for example, the Sultan visited Hebron as part of his religious acts when he visited the shrine of Abraham (*maqām Ibrāhīm*) and visited the highly respected mystic Shaykh ʿAlī al-Bakkā. Baybars took advantage of his visit to Hebron to listen to public complaints (*mazālim*) and solve them. He acted to prepare and present food (*simāṭ al-Khalīl*) to the poor people and even participated in the meals with them. Moreover, the Sultan presented money to Quran readers, imams, attendants, and others. In the same year, Baybars also visited Jerusalem, examining its needs, presenting charity, and attending Friday prayers there. In the year 666/1268, builders and repairmen arrived to Jerusalem to repair the water supply in al-Sulaymāniyya canal to al-Aqsa Mosque (Ibn ʿAbd al-Ẓāhir 1976, pp. 250–51; al-Nuwayrī 2004, vol. 30, p. 96).

### 3.3. Baybars' Interest in Popular Islam: Sufism and Shrines

Baybars took care of developing Islamic education with its institutions in general, including the interest in popular Islam as Sufism and Shrines. His constructions of the madrasa "al-Zahiriyya" in Damascus as well as the "al-Zahiriyya" madrasa in Cairo with their endowments show the extent of Baybars' interest in devoting religious education and its applications (Leiser 1984, pp. 33–55; Holt 1980, pp. 27–35). In his study, Mahamid (2013, pp. 205–14) asserts and concludes that the Mamluk sultan Baybars was known as a supporter of the Sufi order al-Qalandariyya, which came from Syria. Baybars had dedicated a zawiya for its members and provided financial aid and food. Thanks to Baybars' assistance and support, the al-Qalandariyya Sufi Order became stronger, not only in Syria but in Egypt as well. On each of Sultan Baybars' visits to Syria, he donated money, in addition to a yearly allocation of wheat. After Baybars, the Mamluk sultans continued to support Sufi orders and treated them with considerable tolerance, granting them waqf and different aids.

During his travels in the various countries under Mamluk domain, Sultan Baybars showed his interest of popular Islam through his contributions to constructions of shrines and his visits to graves of the righteous. Anne Troadec (2014–2015) claims that "Baybars's

investment in building and restoring monuments during his reign may be seen as a desire to associate himself with key figures of Islamic history". While Frenkel (2001) argues in his study that Baybars tried to legitimize his rule in the eyes of his Muslim people, he concluded that Baybars had achieved three main goals through his Islamization policy: It was as a result of a long mission, establishing his link with the institutionalization of Islam, and fortifying the presence of Islam in the Holy Land. Baybars dealt with generous treatment and a respectful relation with the Sufis and the righteous. He also supported Sufi groups and ascetics, either with money and material aid, or with endowments and building their own institutions, such as *zāwiya*s, *khānqāh*s, and *ribāt*s (Ibn Shaddād 1983, vol. 31, pp. 271–74).

After the victory of Sultan Baybars over the Mongols in the Battle of ʿAyn Jālūt in Palestine, he ordered the construction of a religious shrine in the region in 659/1261 as a sign of victory and as a tribute. Ibn ʿAbd al-Ẓāhir (1976, p. 91) says that this position was honorable, and that it is necessary to know the extent of God's bounty, while al-Nuwayrī (2004, vol. 30, p. 11) mentions this shrine as "*Mashhad al-Naṣr*" (shrine of victory). During Baybars' visits in Syria, he acted to repair and construct tombs of the Prophet's friends (*saḥāba*), such as Khālid ibn al-Walīd's grave in Homs. Baybars ordered to repair it and endow *waqf* for its religious services, as posts of *imām*, *muʾadhdhin* (who calls for prayers), and others. Moreover, he did the same for the grave of Abū ʿUbayda bin al-Jarrāḥ in the Jordan Valley (Ibn Taghrī Birdī 1963, vol. 7, p. 180).

Sultan Baybars used to visit many sites of those graves and shrines that gave special importance to those sites in the views of the people in public spheres. In one of his travels to Syria, he visited the tombs (*maqām*s) of the Prophet's friends (*saḥāba*), such as of Diḥya al-Kalbī in Palestine and of Abū Hurayra in Damascus (Ibn ʿAbd al-Ẓāhir 1976, p. 158). Amitai (2006, pp. 45–53) studied some observations on Baybars' inscription in the Maqam of Nabi Musa between Jerusalem and Jericho, known as the *Maqām Nabī Mūsā* (the shrine of the Prophet Musa), where Baybars was interested in building it by his own order. It is interesting enough to note that Baybars was also visiting the grave of Sheikh ʿAlī bin ʿUlaym, one of the ascetics from the dynasty of ʿUmar ibn al-Khaṭṭāb, and among the owners of dignities and blessings (*al-barakāt wal-karāmāt*) near Arsūf in Palestine (Ibn ʿAbd al-Ẓāhir 1976, pp. 239–42).

Historians of the Middle Ages and the Mamluk era extensively dealt with the charitable works of Sultan Baybars throughout the regions of the Mamluk state in general. It was customary for Baybars to help the poor and the weak people to bring them closer to him with an interest in looking after their state matters, and he was enthralled in the interests of Islamic matters of religion. He gave charity for Sufis in their *zawiya*s and purchased bread for the poor Muslims. Baybars maintained the arrangement of needs and requirements for the orphans of troops, and he even dedicated *waqf* for the shroud of the dead of strangers in Cairo (see: Ibn Taghrī Birdī 1963, vol. 7, pp. 180, 213–14; Ibn ʿAbd al-Ẓāhir 1976, pp. 224–25, 29; Ibn Shaddād 1983, vol. 31, pp. 299–303; al-Maqrīzī 1997, vol. 2, pp. 5–6, 9).

## 4. Conclusions

Baybars succeeded in implementing Islamic principles and reviving Islamic tradition after a period of weakness. He exploited his status as a charismatic Muslim leader with full authority under the restored Caliph, so he acted fiercely to defend Islam and Muslims. In reviewing Sultan Baybars' religious policy and achievements during his reign, it can be concluded that he had achieved a vast range of implementations regarding religious matters, which left their impact on the Mamluk authorities and other Islamic areas in general. The following important achievements in this regard can be highlighted in conclusion:

a.　As a Muslim leader, Sultan Baybars succeeded in reviving the Abbasid Caliphate in Cairo after the conquest of Baghdad by the Mongols in 658/1260, so he deserved relevant titles, such as "Protector of Muslim Countries" (*Ḥāmī bilād al-Muslimīn*), "Partner of the Commander of Faithful People/the Caliph" (*Qasīm Amīr al-Muʾminīn*), and others.

b. Baybars succeeded in introducing judicial reforms to serve the four Islamic schools of thought (*madhāhib*), and officially recognizing them as representatives of Sunni Islam. So, he brought the various judges closer to him, as consultants (*shūra*) in matters of the religion and state. He tied good relations with *ʿulama* of the four Islamic rites, both on the official and the public spheres. Furthermore, he established the "*Dār al-ʿAdl*" (the House of Justice) for implementing Justice in accordance with religion.

c. Baybars' policy of commanding right and forbidding wrong "*al-amr bil-maʿrūf wal-nahy ʿan al-munkar*" was characterized by fighting against innovations (*bidaʿ*) and customs that were contrary to Islamic religion.

d. Baybars' had contributed a lot of charity as *waqf* and religious institutions and acted to control and supervise the holy sites of Islam in Mecca, Medina, al-Quds (Jerusalem), and al-Khalīl (Hebron), and made his strict efforts to provide security, peace, and free worship in these holy places. Thus, he gained the status and title as "the protector of the holy places" (*Ḥāmī al-Ḥaramayn al-Sharīfayn*). He also acted to establish endowments and charity in different places and cities in the mamluk areas for the interest of religious knowledge and sciences, and for popular Islam of Sufism and the holy shrines.

**Funding:** This research received no external funding.

**Institutional Review Board Statement:** The study was conducted in accordance with the Declaration of Helsinki, and approved by the Institutional Review Board (or Ethics Committee) -MDPI Religions Editorial Office.

**Informed Consent Statement:** Not applicable.

**Data Availability Statement:** Not applicable.

**Conflicts of Interest:** The author declares no conflict of interest.

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
