# Peer review of "Religious Policy of the Mamluk Sultan Baybars (1260–1277 AC)"

_religions, doi:10.3390/rel14111384_

Round 1

Reviewer 1 Report

Comments and Suggestions for Authors

It is a very interesting work from a historical point of view. I understand that it sometimes confuses what is Muslim with what is Islamic, but it is something very common in works about the Middle Ages.

Author Response

Response 1

1. Summary

  1. Thank you for taking the time to review my manuscript. I’d like to thank you for your evaluation and comments, and your contribution that helped me make some corrections and re-revision in the required points.
  2. In general, I’ve made the revisions to the whole manuscript as required by highlighting (in red) the new revision, changes, or additions that can be easily reviewed.
  3. The abstract and the introduction have been revised for more understanding to indicate the research methodology. (see the Introduction: lines, 64-72)
  4. References and citations in this article are used according to the style of RELIGION Journal. So, I checked all references and omitted some of them which are unnecessary or irrelevant to the contents of the manuscript, while I’ve added one reference (see lines, 512-513)

Responses and Comments to the Required Points: My explanation of the corrections required in the manuscript and the comments in general, point by point, for your comments:

Comment 3: Are the research design, questions, hypotheses, and methods clearly stated?

After the re-revision and re-review, the research design, questions, hypotheses, and methods are more clearly stated, and focused on the end of the introduction (see lines, 68-87).

Comment 4: Are the arguments and discussion of findings coherent, balanced, and compelling?

The discussion of the research results is coherent, balanced, and convincing, as it relies on qualitative research in terms of producing results through appropriate sources, in the form of search, investigation, comparison while verifying and confronting and presenting the results from medieval and contemporary sources (See the main discussion in the headlines and main topics: lines, 89-90, 149, 202, 278-279, 354, 399).

Comment 7: Are the conclusions thoroughly supported by the results presented in the article or referenced in secondary literature?

As stated in the article and the data discussion, the conclusions are fully supported by the results presented in the article or referred to in the secondary literatur. Anyway, some revisions were made to clarify this point. (see Conclusions: Lines, 456-470).

Reviewer 2 Report

Comments and Suggestions for Authors

Wonky English in places, and largely derivative in scope. No original ideas here.

Comments on the Quality of English Language

Needs improvement

Author Response

Response 2

Summary

a.      Thank you for taking the time to review my manuscript. I’d like to thank you for your evaluation and comments, and your contribution that helped me make some corrections and re-revision in the required points. Special thanks for your positive and lovely comments. Furthermore, I’m attaching below my general responses and explanation of the re-corrections made in the manuscript and the comments in general, point by point.

b.      In general, I’ve made the revisions to the whole manuscript as required by highlighting (in red) the new revision, changes, or additions that can be easily reviewed.

c.      The abstract and the introduction have been revised for more understanding to indicate the research methodology. (see the Introduction: lines, 64-72).

d.     References and citations in this article are used according to the style of RELIGION Journal. So, I checked all references and omitted some of them which are unnecessary or irrelevant to the contents of the manuscript, while I’ve added one reference (see lines, 512-513).

Responses and Comments to the Required Points:

My explanation of the corrections required in the manuscript, point by point in general:

Comment 1: Is the content succinctly described and contextualized with respect to previous and present theoretical background and empirical research (if applicable) on the topic?

The content on the topic was described and summarized within the abstract as well as in the introduction. See repairs, additions, or re-revisions in red. (see lines: 3-8, 12-14, 36-37, 52-53, 62-63, 68-80).

Comment 2: Are all the cited references relevant to the research?

Yes. I’ve checked and omitted some of the irrelevant references to the contents of my manuscript. But I’ve added one more relevant reference (see lines, 512-513)

Comment 3: Are the research design, questions, hypotheses, and methods clearly stated?

After the re-revision and re-review, the research design, questions, hypotheses, and methods are more clearly stated, and focused on the end of the introduction (see lines, 68-87).

Comment 4: Are the arguments and discussion of findings coherent, balanced, and compelling?

The discussion of the research results is coherent, balanced, and convincing, as it relies on qualitative research in terms of producing results through appropriate sources, in the form of search, investigation, comparison while verifying and confronting and presenting the results from medieval and contemporary sources (See the main discussion in the headlines and main topics: lines, 89-90, 149, 202, 278-279, 354, 399).

Comment 5: For empirical research, are the results clearly presented?

I think that this point does not fit with the type and method of this research.

Comment 6: Is the article adequately referenced?

Yes. The article is adequately referenced. I've followed the requirements of the Journal (RELIGION) (see References in the whole article).

Comment 7: Are the conclusions thoroughly supported by the results presented in the article or referenced in secondary literature?

As stated in the article and the data discussion, the conclusions are fully supported by the results presented in the article or referred to in the secondary literatur. Anyway, some revisions were made to clarify this point. (see Conclusions: Lines, 456-470).

Reviewer 3 Report

Comments and Suggestions for Authors

I went through the paper. All I can say is that its English is poor. As you see, I edited the opening section and I left a few comments. The paper should undergo serious proofreading. When it is done and turns out to be a neat paper, send it back to me to check the content. 

Comments on the Quality of English Language

I went through the attached paper, which was sent a few days ago, and I read it twice. All I can say is that its English is poor and disappointing. As you see, I edited the opening section and I left a few comments, both for you and the author, and unless the paper undergoes serious proofreadings, it is not acceptable in this current form. When it is done and turned out to be a neat and acceptable paper, send it back to me to check the content. 

Author Response

Response 3

1. Summary

a.      Thank you for taking the time to review my manuscript. I’d like to thank you for your evaluation and comments, and your contribution that helped me make some corrections and re-revision in the required points.

b.      In general, I’ve made the revisions to the whole manuscript as required by highlighting (in red) the new revision, changes, or additions that can be easily reviewed.

c.      The abstract and the introduction have been revised for more understanding to indicate the research methodology. (see the Introduction: lines, 64-72)

d.     References and citations in this article are used according to the style of RELIGION Journal. So, I checked all references and omitted some of them which are unnecessary or irrelevant to the contents of the manuscript, while I’ve added one reference (see lines, 512-513)

My Responses and Comments to the Required Points:

My explanation of the corrections required in the manuscript, point by point of your comments:

Comment 1: Is the content succinctly described and contextualized with respect to previous and present theoretical background and empirical research (if applicable) on the topic?

The content on the topic was described and summarized within the abstract as well as in the introduction. See repairs, additions, or re-revisions in red. (see lines: 3-8, 12-14, 36-37, 52-53, 62-63, 68-80).

Comment 2: Are all the cited references relevant to the research?

Yes. I’ve checked and omitted some of the irrelevant references to the contents of my manuscript. But I’ve added one more relevant reference (see lines, 512-513)

Comment 3: Are the research design, questions, hypotheses, and methods clearly stated?

After the re-revision and re-review, the research design, questions, hypotheses, and methods are more clearly stated, and focused on the end of the introduction (see lines, 68-87).

Comment 4: Are the arguments and discussion of findings coherent, balanced, and compelling?

The discussion of the research results is coherent, balanced and convincing, as it relies on qualitative research in terms of producing results through appropriate sources, in the form of search, investigation, comparison while verifying and confronting and presenting the results from medieval and contemporary sources (See the main discussion in the headlines and main topics: lines, 89-90, 149, 202, 278-279, 354, 399).

Comment 5: For empirical research, are the results clearly presented?

I think that this point does not fit with the type and method of this research.

Comment 6: Is the article adequately referenced?

Yes. The article is adequately referenced. I've followed the requirements of the Journal (RELIGION) (see References in the whole article).

Comment 7: Are the conclusions thoroughly supported by the results presented in the article or referenced in secondary literature?

As stated in the article and the data discussion, the conclusions are fully supported by the results presented in the article or referred to in the secondary literature. Anyway, some revisions were made to clarify this point. (see Conclusions: Lines, 456-470).

Reviewer 4 Report

Comments and Suggestions for Authors

  1. The abstract must be revised for more understanding, especially the research methodology; concluding remarks and recommendations should be highlighted. 
  2. The study lacks a literature review, which is the backbone of finding out the research gap in the existing literature on the study.
  3. The study also needs to indicate the research methodology to conclude. 
  4. Mumluk's relations with other (non-Muslim) communities of the time must be included. With this part, it can be completed. They especially had what kind of rights and liberties the status of Alh al-Dhimma (which were common in all Muslim States of the time) or any other special citizenship status. 
  5. Mumluk's relations with other states of the Time need to be included.
  6. The language is suitable, but it needs revision for more understanding. 
  7. Some essential recommendations should be drawn separately. 
  8. The reference style also wants correction, especially in the end notes, which need to be revised. 
  9. The Study's Title might be improved with decimation and mentioning the research approach. 
  10. The last one, critical evaluation of the Mamluk's religious policy, should also be part of the study. 
Comments on the Quality of English Language

Overall language is fine. However, Minor editing/proofreading of the English language  is required

Author Response

Response 4

1. Summary

a.      Thank you for taking the time to review my manuscript. I’d like to thank you for your evaluation and comments, and your contribution that helped me make some corrections and re-revision in the required points.

b.      In general, I’ve made the revisions to the whole manuscript as required by highlighting (in red) the new revision, changes, or additions that can be easily reviewed.

c.      The abstract and the introduction have been revised for more understanding to indicate the research methodology. (see the Introduction: lines, 64-72)

d.     References and citations in this article are used according to the style of RELIGION Journal. So, I checked all references and omitted some of them which are unnecessary or irrelevant to the contents of the manuscript, while I’ve added one reference (see lines, 512-513)

Responses and Comments to the Required Points: My explanation of the corrections required in the manuscript and the comments in general, point by point of your comments:

Comment 1: Is the content succinctly described and contextualized with respect to previous and present theoretical background and empirical research (if applicable) on the topic?

The content on the topic was described and summarized within the abstract as well as in the introduction. See re-revisions or additions (in red). (lines: 3-8, 12-14, 36-37, 52-53, 62-63, 68-80).

Comment 5: For empirical research, are the results clearly presented?

I think that this point does not fit with the type and method of this research. (see the Conclusions: lines, 455-483)

Comment 6: Is the article adequately referenced?

Yes. The article is adequately referenced. I've followed the requirements of the Journal (RELIGION) (see References in the whole article).

  1. Summary: On your additional Comments and Suggestions:
  2. I think I’ve dealt with most of the points you mentioned in your Suggestions: (see: Summary 1 above)
  3. Mumluk's relations with other (non-Muslim) communities…: (see the examples of Sultan Baybars on this topic of dealing with the rights of Ahl al-Dhimma. (Lines, 167-174, 198-200). I did not devote a special chapter to dealing with the non-Muslim communities (Ahl al-Dhimma), but the main subject of the research focused on religious and legal policy during the era of Sultan Baybars only.

Round 2

Reviewer 2 Report

Comments and Suggestions for Authors

Accept in its present form.

Comments on the Quality of English Language

-

Reviewer 3 Report

Comments and Suggestions for Authors

The paper has turned out to be fine and is publishable now. 
